# Finding the Time-Period-Based Most Frequent Path from Trajectory–Topology

Jianing Ding 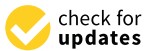, Xin Jin and Zhiheng Li *

Tsinghua Shenzhen International Graduate School, Tsinghua University, Shenzhen 518000, China; djn20@mails.tsinghua.edu.cn (J.D.)

\* Correspondence: zhhli@tsinghua.edu.cn

**Abstract:** The Time-Period-Based Most Frequent Path (TPMFP) problem has been a hot topic in traffic studies for many years. The TPMFP problem involves finding the most frequent path between two locations by observing the travelling behaviors of drivers in a specific time period. However, the previous researchers over-simplify the road network, which results in the ignorance of transfer costs at intersections. To address this problem more elegantly, we built up an urban topology model consisting of Intersection Vertices and Connection Vertices. Specifically, we split the Intersection Vertices to eliminate the influence of transfer cost on finding TPMFP and generate Trajectory–Topology from GPS records data. In addition, we further leveraged the Footmark Graph method to find the TPMFP. Finally, we conducted extensive experiments using a real-world dataset containing over eight million GPS records. Compared to the current state-of-the-art method, our proposed approach can find more reasonable MFP in approximately 10% of cases during off-peak hours and 40% of cases during peak hours.

**Keywords:** path finding; urban topology modeling; big trajectory data

## 1. Introduction

The Time-Period-Based Most Frequent Path (TPMFP) problem has been a hot topic in traffic studies for many years. Generally speaking, TPMFP is the most commonly traversed path in a directed network during a specific period. The study of the most frequent path (MFP) is motivated by the increasing need to analyze and understand complex networks. With the emergence of large-scale datasets in various domains, such as social networks [1], transportation systems [2], and web browsing behavior [3], there is a growing demand for efficient algorithms to extract meaningful information from these networks. MFP provides a powerful tool for identifying the most commonly traversed paths in a directed network, which can have important implications for network optimization, anomaly detection, and targeted interventions. As a result, MFP has become a popular research topic in fields such as data mining and network analysis.

In traffic research, the frequent path refers to the path frequently passed by the moving object. It can be a complete and actual road or a collection of several sections that are not completely connected. Since the shortest or fastest path is not always the best, the purpose of studying popular routes is to find the most popular route between two places by studying the behavior of other drivers. In the map service and vehicle navigation system, TPMFP provides users with additional path options besides the shortest/fastest path. For example, when people travel in an unfamiliar city, they tend to take the most common route to avoid getting lost and encountering unexpected congestion. In these cases, TPMFP is better than the shortest/fastest path.

The typical approach used in related research is to use statistical methods that model the frequency distribution of paths and estimate the most probable ones. For example, a straightforward method for finding MFP in the road network is to count the trajectories

going through the path and select the path with the highest support [2]. Figure 1 gives an illustrative example, showing three $V_1 - V_5$ paths with non-zero support, and the paths traversed by $G_1, G_2, G_3$ whose supports are 8, 6 and 4, respectively. Thus, the most frequent path is $G_1$ ($V_1 \rightarrow V_2 \rightarrow V_4 \rightarrow V_5$), which has the most support. However, this method only makes a comparison of path frequencies, ignoring the frequency on road sections. Under this judgment, we cannot infer that the most frequent path from $V_2$ to $V_5$ is $V_2 \rightarrow V_4 \rightarrow V_5$, because path $V_2 \rightarrow V_3 \rightarrow V_5$ has more supports in total. In other words, the path with the highest support is not always suffix-optimal.

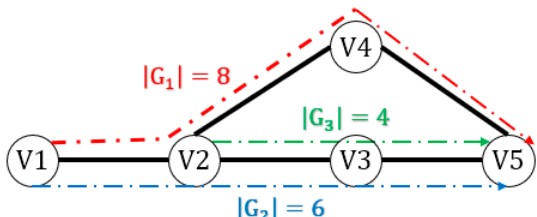

**Figure 1.** An illustrative example of the most frequent path.

Ref. [4] proposed a new method for addressing this issue, which used an ascending frequency sequence to define the path frequency and considered the size of the smallest element in the sequence to determine the path frequency. A Footmark Graph was created to calculate the frequency of roads according to trajectory data. In the example of Figure 1, the frequencies of roads in path $V_1 \rightarrow V_2 \rightarrow V_4 \rightarrow V_5$ were 14, 8 and 8, respectively, so the path frequency was (8, 8, 14), while the frequency of path $V_1 \rightarrow V_2 \rightarrow V_3 \rightarrow V_5$ was (10, 10, 14). Therefore, the latter path was more frequent.

Intuitively, in this case, 57% of the drivers from $V_1$ to $V_5$ choose route $V_1 \rightarrow V_2 \rightarrow V_4 \rightarrow V_5$, which seemed to be a better choice for most drivers. There has yet to be further discussion on this phenomenon in [4]. Ref. [5] used a threshold $\theta$ to make the method that finds the most frequent path more reasonable. According to the improved method, an ascending frequency sequence was more frequent than another only if the D-value of their smallest elements was bigger than $\theta$.

However, this method does not analyze the underlying causes of this phenomenon. We supposed that the road network had been over-simplified in past research, so that the drivers' tendencies to shift in different directions at the intersection had been ignored. Figure 2 gives an example. Due to the Suffix-optimal principle, $V_1 \rightarrow V_2 \rightarrow V_4 \rightarrow V_6$ is the MFP of $V_1 \rightarrow V_6$. However, the reason why the five drivers (the red line) did not choose MFP is probably that the transfer cost of path $V_1 \rightarrow V_2 \rightarrow V_4$ at $V_2$ was far greater than that of $V_1 \rightarrow V_2 \rightarrow V_5$, leading to the drivers from $V_1$ preferring to go straight at $V_2$ rather than turning left. This phenomenon is widespread in the actual road network when road $V_3 \rightarrow V_2 \rightarrow V_4$ is a main road with large traffic flow [6–8].

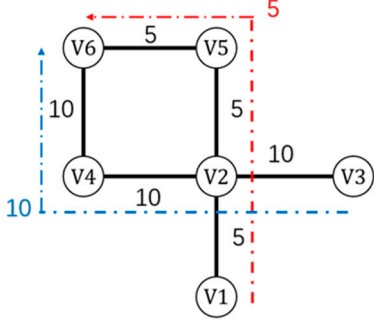

**Figure 2.** An example of drivers' tendencies to choose proper directions. Ten vehicles pass through the blue path, and five vehicles pass through the red path.

Trajectory big data represent an important data source for studying TPMFP problems, coming from the real-time positioning data reported by the floating car [9]. High-quality

trajectory data are crucial to the research based on trajectory data [10–12]. Map matching is the key step in obtaining trajectory data from floating car data. Map matching is the process of aligning the observed position sequence with the road network. Because the urban space is continuous, the expression domain of trajectory points is infinite [13]. Therefore, it is necessary to discretize the urban road network before map matching. The common method in past research has been to match the trajectory points to the road sections [14]. However, due to the low topological degree of urban space, the map-matching process is time-consuming [15,16], and the expression form of the trajectory is complex. Another simple method is to model the urban road network as a directed graph $(V, E)$, where $V$ is the collection of all intersections, and then match all global positioning system (GPS) records to the intersection points and represent the trajectory as the connection of the intersections [17]. This method brings a high degree of simplification to the road network. As a result, the obtained trajectory data lose helpful information to a certain degree.

A precise topology model could help reduce computational complexity and keep most helpful information in such studies. For example, ref. [18] presents a clustering point process (CPP)-based network topology structure. The feature points, such as endpoints, bends, and crossroads in a road system, are connected by lines to characterize the network structure. This topology model can support the generation of the road extraction model and the operation of the extraction algorithm. Ref. [19] extracted the road topology from the street layout of Shanghai city, and made a modification to make the road network topology analogous to the real-world road network. The resulting road topology was imported to Simulation of Urban Mobility (SUMO) to characterize different aspects of vehicular mobility.

In this paper, we studied the problem of TPMFP in trajectory data and a topology model of the urban road network. We extracted all meaningful trips from the original GPS data, and matched the GPS points with the topology model. The resulting series of connected vertices is called Trajectory–Topology. In order to study the impact of intersections' transfer costs on frequent paths, we divided the intersections into multiple virtual connection vertices connected to each other. With the help of the precise urban topology model, we represented the trip paths as a series of adjacent connection vertices and re-studied the TPMFP problem using the Footmark method. The main contributions of this paper are as follows:

- We designed a precise urban topology model (Topo Map) to express trajectory data. The original GPS data are converted into Trajectory–Topology;
- We studied the impact of intersections' transfer costs on frequent paths by finding TPMFP from the Trajectory–Topology and Topo Map, verifying the validity of the suffix-optimal principle in the TPMFP problem;
- We conducted extensive experiments using a real dataset containing over eight million GPS records to evaluate the performance of our method. The results showed that our approaches found more reasonable frequent paths than state-of-the-art baselines.

## 2. Preliminary

### 2.1. Data Description

Our GPS records data set was provided by Hangzhou Transportation Satellite Positioning Application Co., Ltd., Hangzhou, China. The data set had 8,507,317 GPS records on 8 January 2021, covering 31,235 electric and hybrid taxis driving in Hangzhou. Three types of information, space information, time information and vehicle status information, were shown in the data set. Each GPS record consisted of 6 fields, as described in Table 1.

The road network file we used to build the urban topology model was the road network of the core area of Hangzhou, including Shangcheng District, Xiacheng District, Jianggann District, Gongshu District, West Lake District and Binjiang District. These six districts are the core business district of modern Hangzhou, with a dense road network and large traffic flow, which is convenient for obtaining rich data. Figure 3 gives an overview of

the GPS records data we used for research. The sampling points were spread all over our research area.

**Table 1.** List of GPS records data fields.

| Field | Description |
|---|---|
| Vin | Unique identification of a vehicle |
| Longitude | The longitude of GPS record |
| Latitude | The latitude of GPS record |
| Collect Time | The time when GPS record was recorded |
| Speed | The speed of vehicle |
| Vehicle State | The state of the vehicle (for hire or not) |

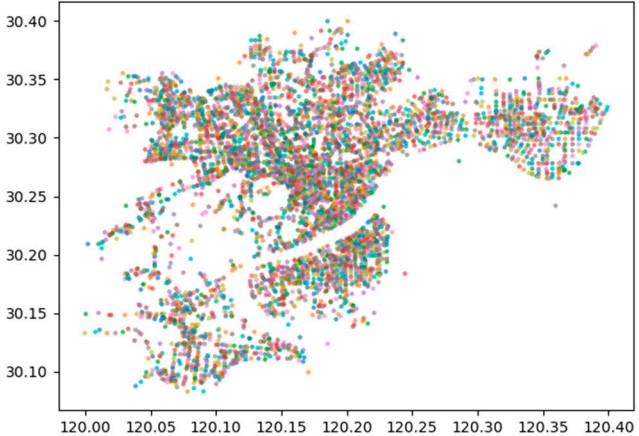

**Figure 3.** An overview of the GPS records data.

In order to build a precise urban topology model, we discretized the urban space into a topological graph composed of several vertices with semantic elements connected to each other. Firstly, we extracted all the intersections of the road network by extracting the intersections of road center lines. The vertices we obtained were defined as Intersection Vertices. Secondly, with the help of the Gaode open website, one of China's greatest digital map providers, 2785 points of interest (POI) in the core area of Hangzhou were collected. We projected those POI onto the road network and defined the projection points as the Connection Vertices. Finally, the Connection Vertices and Intersection Vertices were connected according to their connection relationship in the road network. In this way, an urban topology model was established. Our topology model had 4211 Intersection Vertices and 2758 Connection Vertices, connected by 9055 edges. We called this model Topo Map. Figure 4 shows a part of the Topo Map.

*2.2. Problem Statement*

The Time-Period-Based Most Frequent Path problem is a time-period-based map query problem. Given a time period $T$, a source $V_s$ and a destination $V_d$, TPMFP searches the MFP from $V_s$ to $V_d$ during $T$.

Before giving the formal definition of TPMFP, some essential related definitions are given below.

**Definition 1.** *(Feature vertices) Feature vertices are a set $V = \{I, C\}$, where $I$ is the set of Intersection Vertices, and $C$ is the set of Connection Vertices.*

**Definition 2.** *(Topo Map) A Topo Map is a directed graph $G = (V, E)$ where $V$ is the set of feature vertices, and $E$ represents an adjacency matrix according to the roads.*

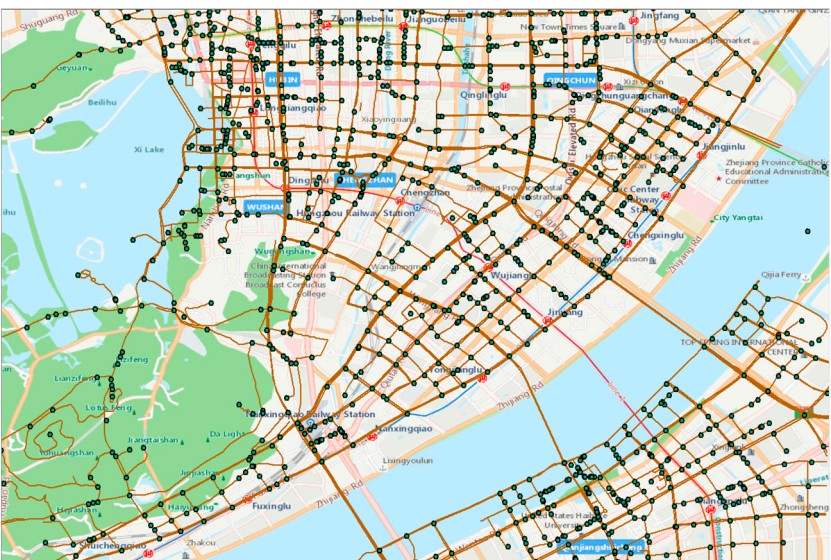

**Figure 4.** Part of the Topo Map. Each green dot represents an Intersection Vertex or a Connection Vertex.

**Definition 3.** *(Path) Given G, an $x_1 - x_k$ path is a nonempty graph $P = (V_p, E_p)$ of the form $V_p = \{x_1, x_2, \dots x_k\}$ and $E_p = \{(x_1, x_2), (x_2, x_3) \dots (x_{k-1}, x_k)\}$ such that P is a sub-graph of G and the $x_i$ are all distinct.*

**Definition 4.** *(Trajectory–Topology) Given G, a Trajectory–Topology Y is a feature vertices sequence $(x_1, t_1), (x_2, t_2), \dots (x_k, t_k)$ formed by an individual passing through the feature vertices in turn from time $t_1$ to $t_k$, where $t_i$ represents the time that the location of the individual is matched to feature vertex $x_i$.*

**Definition 5.** *(Footmark) Given $\Omega = (G, V_d, Y, T)$, $V_d$ represents the destination vertex in Top Map and T is the time period. For each topological trajectory $Y_i$ in Y, if there exists a non-empty sub-trajectory $Y_i' = ((x_1, t_1), (x_2, t_2), \dots (x_k, t_k))$ of $Y_i$ such that:*

- $[t_1, t_k] \subseteq T$.
- $x_k = V_d$.

Then, for each $x_{i-1} - x_i$ Path in sub-trajectory $Y_i'$, the edge $(x_{i-1}, x_i) \subseteq G$ receives a count of Footmark.

**Definition 6.** *(Edge Frequency) Given $\Omega = (G, V_d, Y, T)$, the Edge Frequency $f(u, v)$ of $(u, v) \subseteq G$ is the number of the Footmarks in $(u, v)$.*

**Definition 7.** *(Footmark Graph) Given $\Omega = (G, V_d, Y, T)$, a Footmark Graph $G_f$ is a sub-graph of G ; for each $(u, v) \subseteq G_f$, there is $f(u, v) > 0$.*

**Definition 8.** *(Path Frequency) Given $G_f$, the Path Frequency $F(P)$ of an $x_s - x_d$ Path is a sequence sorted from smallest to largest, whose elements are the Edge Frequency of the edges in the Path.*

**Definition 9.** *(More-Frequent-Than) Given two Path Frequencies $F(P_1)$ and $F(P_2)$, Path $P_1$ is more frequent than Path $P_2$ only if any of the following statements holds, denoted as $F(P_1) \geq F(P_2)$:*

- *$F(P_1)$ is prefix of $F(P_2)$.*
- *The lexicographic order is $F(P_1)$ is bigger than $F(P_2)$.*

**Definition 10.** (MFP) *Given $G_f$ and a $V_s - V_d$ Path $P^* \subseteq G_f$, $P^*$ is the MFP with respect to $G_f$ only if for any $V_s - V_d$ Path $P \subseteq G_f$, there is $F(P^*) \geq F(P)$.*

Finally, the TPMFP problem can be stated as: given $\Omega = (G, V_s, V_d, Y, T)$, we need to find the MFP from $V_s$ to $V_d$.

### 3. Trajectory–Topology Generation

*3.1. Improved Interactive-Voting-Based Map Matching Method*

The original GPS data were a series of non-uniform sampling points. Thus, before analyzing trajectory data, the GPS points had to be matched to the road network to reveal the hidden semantic information. In our work, an improved Interactive-Voting-Based Map Matching (IVMM) [20] method was taken into account to match the GPS points to the Topo Map.

IVMM is a global algorithm that employs a voting-based approach to reflect the mutual influence of the sampling points. This algorithm applies to scenarios with low-sampling-rate data. Ref. [20] proved that the correct matching percentage could reach 70% when the sampling interval of GPS records was 4.5 min. The characteristic of low-sampling-rate GPS records data was that two adjacent points in a time sequence may be separated by several road sections. In our dataset, the average sampling interval of GPS data was 30 s, which belonged to the dense-sampling-rate data [21–23]. However, it should be noted that the density of vertices in the Topo Map was higher than that of the road network, which consists of intersections and road sections. As a result, two points that are not far apart in the road network may be separated by several feature vertices in our topology model. This condition is similar to matching low-sampling-rate data to the road network. Hence, we used IVMM to generate Trajectory–Topology. Parts of the algorithm have been modified to adapt to the topology model. The matching process can be divided into four steps. Firstly, a candidate set was built up for each sampling point. Let $p_i$ be a sampling point, and $CP_i$ denotes the candidate set of $p_i$. Set $CP_i$ contains all the vertices in the Topo Map within a radius $r$ (a fixed number) of $p_i$ with respect to Euclidian distance, as shown in Figure 5.

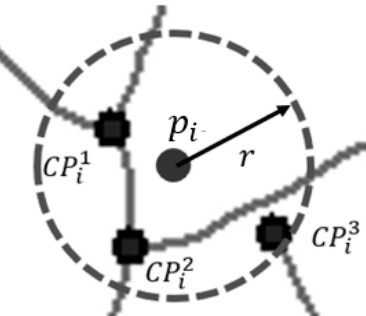

**Figure 5.** An example of a sampling point and its candidate set.

Second were position and topology analysis in conduct. We assume that the distribution of the measurement error of the sampling points satisfies the Gaussian distribution $N(\mu, \sigma^2)$ [24]. Thus, the error rate of a candidate vertex $CP_i^j$ with respect to $p_i$ is formulated as:

$$\epsilon\left(CP_i^j\right) = exp\left(-\frac{\left(x_i^j - \mu\right)^2}{2\sigma^2}\right) \qquad (1)$$

where $x_i^j$ is the Euclidian distance from $CP_i^j$ to $p_i$. Then, we defined $V\left(CP_{i-1}^j \to CP_i^k\right)$ as the transition probability from the candidate vertex $CP_{i-1}^j$ to $CP_i^k$:

$$V\left(CP_{i-1}^j \to CP_i^k\right) = \frac{Eu\_dist(p_i, p_{i-1})}{Topo\_dist\left(CP_{i-1}^j, CP_i^k\right)} \tag{2}$$

where $Eu\_dist(p_i, p_{i-1})$ is the Euclidian distance from $p_{i-1}$ to $p_i$, and $Topo\_dist\left(CP_{i-1}^j, CP_i^k\right)$ is the shortest path length from $CP_{i-1}^j$ to $CP_i^k$ in the Topo Map. It is worth mentioning that the shortest path length of two vertices in the Topo Map could be pre-computed by Floyd's Algorithm [25] so as to reduce the calculation cost when matching. Notice that there are fewer than 7000 vertices in the Topo Map, and the running time of Floyd's algorithm was acceptable. After the calculation of transition probability, the spatial analysis function was formulated as follows:

$$F_s\left(CP_{i-1}^j \to CP_i^k\right) = \epsilon\left(CP_i^k\right) \times V\left(CP_{i-1}^j \to CP_i^k\right) \tag{3}$$

Thirdly, Mutual Influence Modeling was conducted. Define $M = diag\left\{M^{(2)}, M^{(3)}, \ldots, M^{(n)}\right\}$, while $M^{(i)}$ is the spatial analysis function matrix of $CP_{i-1}$ and $CP_i$. We used the negative exponential function as the distance weight function:

$$w_i^{(j)} = exp\left(\frac{-Eu\_dist(p_i, p_{i-1})^2}{\beta^2}\right) \tag{4}$$

where $\beta$ is a fixed value. Next, the Distance Weight Matrix $W_i$ was defined as $W_i = diag\left\{W_i^{(j)}\right\}$, $j \neq i$. Matrix $W_i$ gives weights for the effect of all other points to $p_i$ associated with their distances to $p_i$. After that, the Weighted Score Matrix was formulated as:

$$\phi_i = W_i M \tag{5}$$

Matrix $\phi_i$ gives the similarity of all the candidate feature vertices with the actual vertices sequence on the Topo Map, taking the influence of the distance into consideration.

Finally, for all candidate vertices of each sampling point, it is assumed that they are the correct matching vertices of the corresponding sampling points, then the dynamic programming algorithm is used to find an optimal path through the candidate points and count the votes for all candidate points passing through by the optimal path once. Finally, the candidate vertices with the highest cumulative votes in each candidate set are connected to form a Trajectory–Topology. The algorithm in pseudo-code is presented as follows (Algorithm 1):

---

**Algorithm 1:** Interactive Voting

---

Input: $M$, candidate set $CP$:
Output: The vote counts for each candidate vertex
1: **for** $1 \to i$ **to** $n$ **do**
2:    Compute $\phi_i$ and $W_i$
3:    Pre_Path = FindPreNode($CP$, $\phi_i$)
4:    Next_Path = FindNextNode($CP$, $\phi_i$)
5:    **for** $1 \to k$ **to** len($CP_i$) **do**
6:     Vote for each vertex in Pre_Path and Next_Path

---

Given the candidate set and $\phi_i$, the FindPreNode and FindNextNode processes are called to find the optimal path that ends and starts with each vertex in $CP_i$. The two processes are called only once for each candidate set, assuming that the average number of

elements in the candidate set is $k$, the time complexity of each process is $O(nk^2)$, while the time complexity of voting is $O(nk)$. Notice that the calculation process of each sampling point is independent. Thus, the time complexity of Algorithm 1 can be optimized to $O(nk^2)$ by using parallel computing.

The vertices we derived after voting were connected by the sequence of the sampling points and were defined as Trajectory–Topology in our work. Figure 6 shows an example of Trajectory–Topology. The red lines are the original sampling points' segments, while the blue lines are the corresponding Trajectory–Topology. The non-uniform sampling points are effectively represented as a feature vertices sequence.

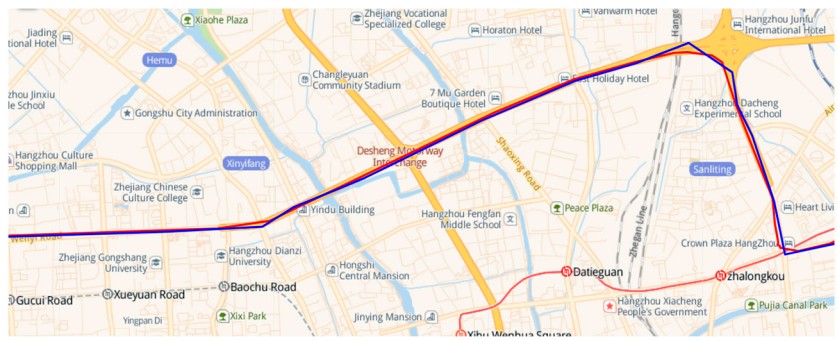

**Figure 6.** Original sampling points (red lines) and corresponding Trajectory–Topology (blue lines).

### 3.2. Trajectory–Topology Smoothing

The work described in Section 3.1 can match the original sampling points to the corresponding feature vertices in the Topo Map. However, in practical applications, it is necessary to know not only which feature vertex corresponds to a sampling point in the map, but also the feature vertices and edges in the Topo Map that the Trajectory–Topology passes through. Therefore, the matched vertices were usually connected in turn in the Topo Map. In other words, Trajectory–Topology Smoothing is used to find a path that exists in the topology map, and the path passes through all nodes of the original Trajectory–Topology.

Figure 7 shows an example of a Trajectory–Topology that has not been smoothed. It can be inferred that the vehicle passed an intersection and turned. However, due to the sampling rate, the intersection is non-existent in both the original sampling points and the corresponding Trajectory–Topology. As a result, the Intersection Vertex would be ignored when building up the Footmark Graph. On the other hand, the Connection Vertices are densely distributed on some road sections. Thus, two adjacent points of a Trajectory–Topology may be separated by several Connection Vertices, which results in a large amount of information loss.

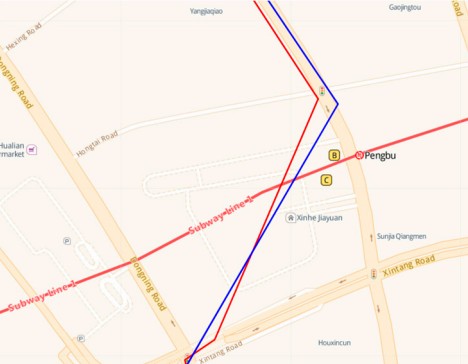

**Figure 7.** The original GPS trajectory (red lines) and the corresponding Trajectory–Topology that has not been smoothed (blue lines).

In our work, we used the Shortest Path to smooth the Trajectory–Topology. For each vertex in a Trajectory–Topology, if it was not directly connected to its last vertex in the Topo Map, the Shortest-Path that consisted of serval vertices between the two vertices would be calculated and added to the Smoothed Trajectory–Topology, denoted as $SY$. Owing to the 30 s sampling rate of our data, using the Shortest-Path was an effective way to complete the path between two vertices. The algorithm in pseudo-code is presented as follows (Algorithm 2):

---

**Algorithm 2:** Trajectory–Topology Smoothing

---

Input: Trajectory–Topology $Y$:
Output: Smoothed Trajectory–Topology $SY$
1: $SY$ = List
2: Add the first vertex of $Y$ to $SY$
3: $1 \rightarrow m$
4: **for** $1 \rightarrow i$ **to** $n$ **do**
5:     **if** $Y_2.vertex$ is directly connected to $SY_m.vertex$
6:         Add $Y_2$ to $SY$
7:     **else**
8:         Path = FindShortestPath($SY_m.vertex$, $Y_2.vertex$)
9:         for vertex in Path
10:            Add vertex to $SY$
11:    $m + 1 \rightarrow m$
12: **return** $SY$

---

In Section 3.1, the shortest path length of each pair of vertices was calculated. According to the calculated length, we used Breadth First Search to find the specific path between two vertices. In particular, the FindShortestPath process extends outward from the source vertex and discards paths whose current distance exceeds the shortest path length until the target point is found. If the shortest path consists of $m$ vertices on average, the time complexity of the process is $O(m)$ in most cases. It may reach $O(m^2)$ when Intersection Vertices are densely distributed. The time complexity of Algorithm 2 is $O(mn)$ on average.

Figure 8 gives an example of Trajectory–Topology Smoothing. The blue lines are the Smoothed Trajectory–Topology, while the red lines are the original ones. The original broken line formed by connecting vertices far away from each other is transformed into a smooth line formed by connecting adjacent vertices. Hence, all information on the path through which Trajectory–Topology passes could be derived.

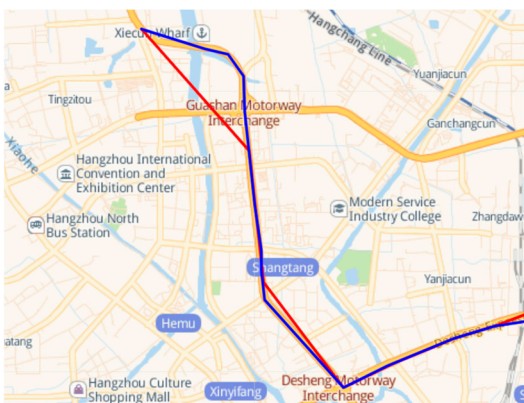

**Figure 8.** Smoothed Trajectory–Topology (blue lines) and the original one (red lines).

## 4. Intersection Vertices Split

An Intersection Vertex usually means vehicles have a waiting cost when passing through. In the Figure 2 example, we assumed that the transfer cost at Intersection Vertex $V_2$ of Path $V_1 \rightarrow V_2 \rightarrow V_4$ was the main reason that drivers from $V_1$ did not choose this path.

In this section, we re-model the Intersection Vertices by splitting them into several Virtual Connection Vertices and re-establish the connection with their adjacent vertices, to eliminate the influence of transfer cost in intersections when searching the most frequent path.

Figure 9 demonstrates how the Intersection Vertices Split is conducted. Firstly, the adjacent Connection Vertices (denoted as $C_i$) of an Intersection Vertex ($I$) are collected. Secondly, the Intersection is split into the same number of its adjacent vertices, denoted as Virtual Connection Vertices ($VC$). For example, in Figure 9, there are four adjacent vertices of $I_1$, so the number of corresponding $VC$ is four. Finally, the four $VC$ are connected to each other and make a connection with their adjacent vertices, in a replacement of the original Intersection.

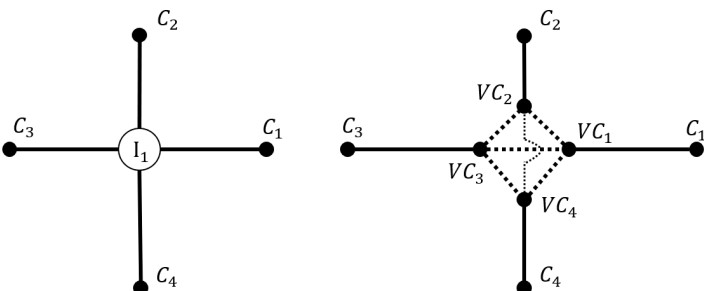

**Figure 9.** Intersection Vertices Split in Topo Map.

Figure 10 illustrates why the Intersection Vertices Split can eliminate the influence of transfer cost in intersections, given Trajectory–Topology $Y$ whose vertices sequence is $C_1 \rightarrow I_1 \rightarrow C_3 \rightarrow I_2$ after Trajectory–Topology Smoothing. Without the Intersection Vertices Split, the $I_1 \rightarrow C_3$ edge and $C_3 \rightarrow I_2$ will receive a Footmark, respectively, when building up the Footmark Graph whose destination Vertex is $I_2$. That results in the rise in the Path Frequency of Path $C_4 \rightarrow I_1 \rightarrow C_3 \rightarrow I_2$ when comparing with Path $C_4 \rightarrow I_1 \rightarrow C_2 \rightarrow I_2$. After splitting $I_1$, the Path $C_4 \rightarrow I_1 \rightarrow C_3 \rightarrow I_2$ becomes $C_4 \rightarrow VC_4 \rightarrow VC_3 \rightarrow C_3 \rightarrow I_2$, denoted as $P_1$. The number of vehicles from $C_1$ will not increase the Edge Frequency of $VC_4 \rightarrow VC_3$, which represents the willingness of drivers from $C_4$ to change the direction to $C_3$ at Intersection $I_1$. According to the topological relationship, the Edge Frequency of $VC_4 \rightarrow VC_3$ is less than or equal to that of $VC_3 \rightarrow C_3$. As a result, the smallest element of $F(P_1)$ does not increase. It can be inferred that given a $C_4 \rightarrow I_2$ Path $P^*$, while $F(P^*)$ is strictly larger than $F(P_1)$, no matter how many vehicles pass the route $C_1 \rightarrow I_1 \rightarrow C_3 \rightarrow I_2$, $F(P^*)$ is still strictly larger than $F(P_1)$ after the Intersection Vertices Split. This method eliminates the influence of vehicles in the other direction on the driver's direction selection at a specific Intersection Vertex when finding the MFP.

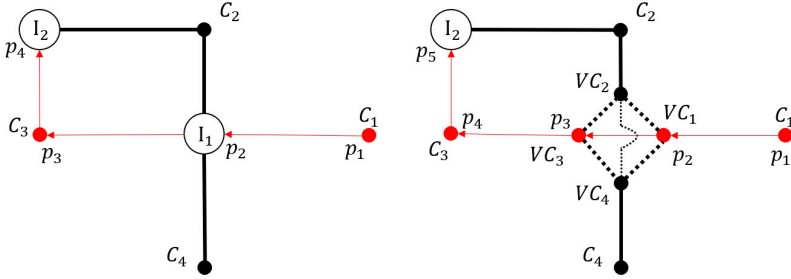

**Figure 10.** The form of Trajectory–Topology after the Intersection Vertices Split (red lines).

## 5. Method of Searching TPMFP

Given $(G_f, V_s, V_d)$, the task of Searching TPMFP is to find a $V_s - V_d$ Path with the highest Path Frequency. In our work, we used a dynamic programming algorithm to find

the MFP between two given vertices. Let $F^*(V_s)$ be the frequency of the $V_s - V_d$ MFP. Then, the corresponding state transition equation is:

$$F^*(V_s) = \max\{f(V_s, v) + F^*(v)\} \tag{6}$$

where $v$ is the adjacent vertex of $V_s$. According to the state transition equation, the MFP problem is similar to the shortest path problem of a single source point, which can be solved by Dijkstra's Algorithm [26].

We used a binary heap to optimize the calculation process. Algorithm 3 calculates the $V_s - V_d$ MFP in time $O\left(\left|E_f\right| \log \left|V_f\right|\right)$, where $\left|E_f\right|$ and $\left|V_f\right|$ are the numbers of edges and vertices in $G_f$. Considering that the Topo Map is a Sparse graph, this method could produce non-negligible optimization in time complexity.

---

**Algorithm 3: MFP**

---

Input: $V_s, G_f, V_d$
Output: $V_s - V_d$ MFP
1: *Nodes* = Heap
2: Add $V_d$ to *Nodes*
3: *None* $\to F^*(V)$ for $V$ in $G_f$
4: **while** *Nodes* is not none **do**:
5:      pop up the node with highest $F^*$ value in *Nodes* $\to v$
6:      if $v$ has been marked: **continue**
7:      mark $v$
8:          **for** each node $u$ adjacent to $v$ and $u$ has not been marked **do**
9:              **if** $f(u, v) + F^*(v) \geq F^*(u)$ **then**
10:                 $u.next = v$
11:                 $F^*(u) = f(u, v) + F^*(v)$
12:                 Add $u$ to *Nodes*
13: create $P^*$ by following the next vertex from $V_s$ to $V_d$
14: **return** $P^*$

---

## 6. Experiment

In this section, we used the dataset described in Section 2.1 to conduct the experiments.

### 6.1. Data Processing

Since taxis have the characteristics of searching and carrying passengers, a complete taxi trajectory usually cannot accurately reflect the driver's route choice between two given points [27–29]. Therefore, we extracted 16,142 trips from the data according to the Vehicle State field, which represented whether there was a passenger in the taxi. We used IVMM to generate Trajectory–Topologies and conduct Smoothing. The derived Trajectory–Topologies are shown in Figure 11a. We counted the top 100 hottest vertices in the Topo Map, as shown in Figure 11b, where the red marker represents an Intersection Vertex and the green marker represents a Connection Vertex. It can be seen that a large proportion of the markers were distributed on two main roads. Most of the hottest vertices were Intersections because they undertake traffic flow in different directions.

### 6.2. Results

In order to evaluate the effectiveness, we compared the results of TPMFP with Intersection Vertices Split (MFP(s)), TPMFP without Split (MFP) and the shortest path (STP). First, the smoothed Trajectory–Topologies and the state-of-the-art method proposed in [4] were used to find the TPMFP without Split. Then, the Trajectory–Topologies were re-smoothed after splitting the Intersections to find the TPMFP with Split, using the method proposed in Section 5. Finally, the STP was calculated using the shortest path algorithm.

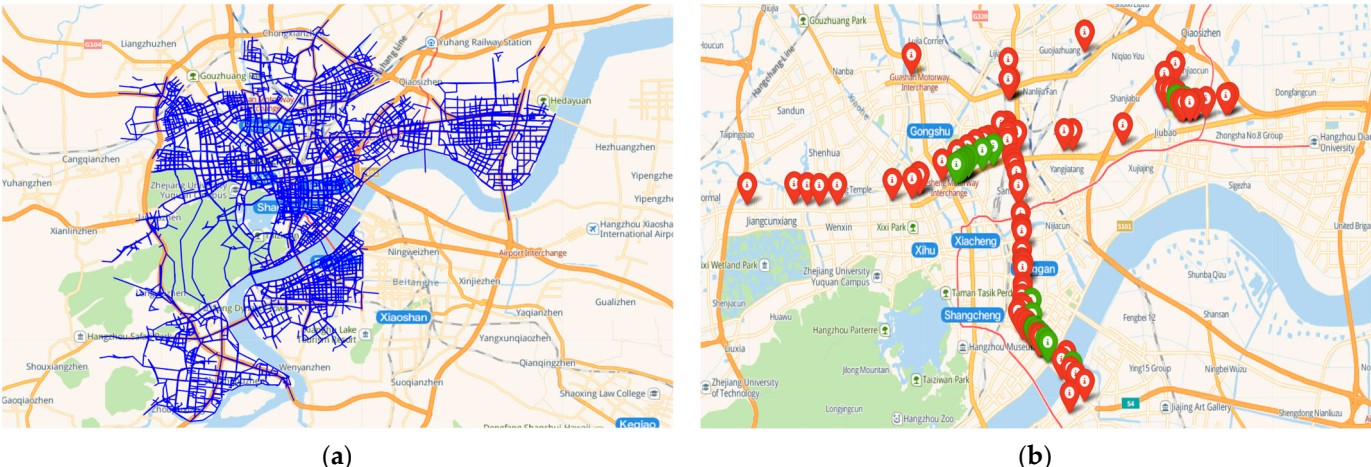

**Figure 11.** A screenshot of the data after processing: (**a**) distribution of the Smoothed Trajectory–Topologies; (**b**) the top 100 hottest vertices in Topo Map, each red marker represents an Intersection Vertex and each green marker represents a Connection Vertex.

The question that we were curious about is whether the MFP was different after Intersection Vertices Split. We chose two vertices on both sides of the east–west road, shown in Figure 11b, and studied the MFP(s) and MFP in rush hours (17:00–19:00) and off-peak hours (11:00–13:00). We used Dijkstra's Algorithm to find the STP and mark it on the figure. According to Figure 12a, a different $V_s - V_d$ MFP (the blue lines) was found after splitting. The MFP and MFP(s) diverged when they reached intersection $I_1$ with a large flow in the east–west direction in the rush hours. This phenomenon can be explained by the difference in traffic capacity in different directions of the intersection [30,31]. Main roads usually have greater right-of-way, so the drivers from the east tended to go straight while driving to $V_d$ in the Figure 12 example. These trajectories contribute to the frequency of the edges in MFP. As a result, the MFP chose to turn left at $I_1$ according to the suffix-optimal principle [11]. However, the MFP(s) showed that drivers from $V_s$ prefer going straight at $I_1$ to avoid busy sections, which seems to be a more reasonable choice because the waiting time at the left turn direction at $I_1$ is possibly long [32].

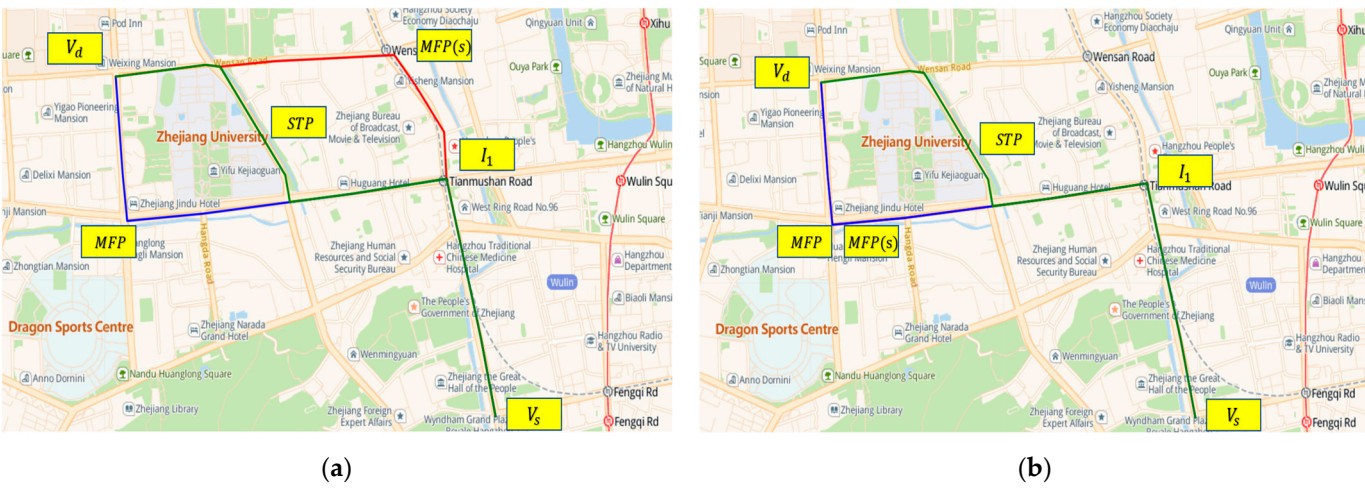

**Figure 12.** Example of MFP, MFP(s), and STP of different time periods. (**a**) The example at rush hours; (**b**) the example at off-peak hours.

In Figure 12b, the MFP and MFP(s) became identical. That can be explained by the drivers' preference to choose the main road with better road conditions when there is no congestion [33,34]. Notice that the STP was neither MFP nor MFP(s). It can be inferred that

drivers are more willing to reduce the number of turns when the distance of two paths is close.

To further verify our hypothesis, we chose the hottest Intersection Vertex as the destination and the other Intersections as the source. There were 2785 Intersection Vertices in the Topo Map, and we derived 2474 MFP and MFP(s) in off-peak hours and 2531 in rush hours since there may have been no trajectories between two vertices in a specific time period. According to Figure 13a, only about 10% of the MFP(s) were different from MFP in off-peak hours. However, that number rose to about 40% when it came to the rush-hours, as shown in Figure 13b. MFP(s) was not equal to MFP, meaning that the transfer cost at intersections caused MFP to choose an incorrect path, as in the example provided in Figure 12. This result further validates our assumption that the transfer cost at intersections influences the finding of the MFP. The direction selection at intersections of an MFP may be misled by the traffic flow on the road in the other direction, especially during rush hours. The fundamental reason for this phenomenon is that the oversimplified road network model makes researchers ignore the waiting cost of vertices. The comparison results demonstrate that our method can find more reasonable MFP during rush hours.

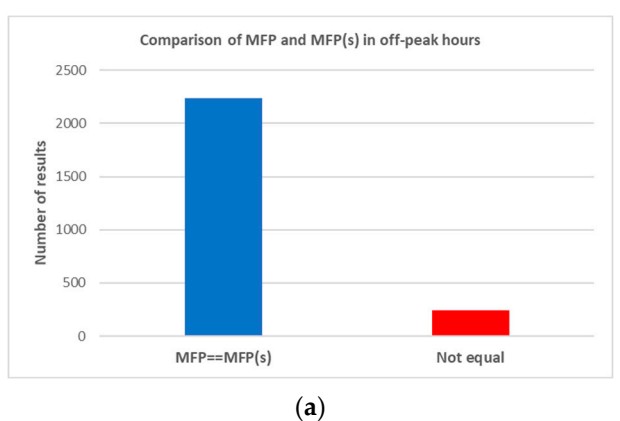
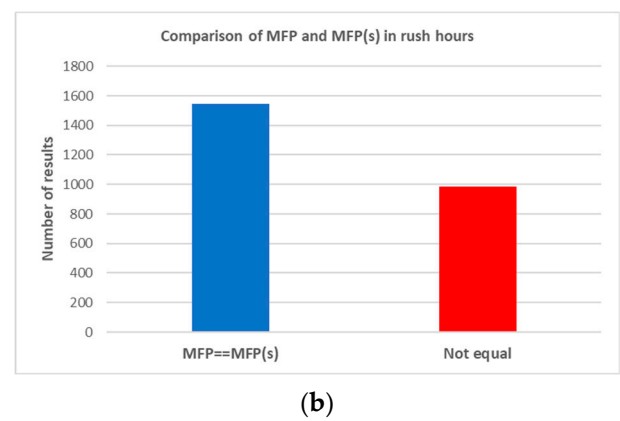

(**a**)　　　　　　　　　　　　　　　　　　　(**b**)

**Figure 13.** Statistics of query result. (**a**) The result in off-peak hours; (**b**) the result in rush hours.

## 7. Conclusions

In this paper, we studied the problem of finding the time-period-based most frequent path based on Trajectory–Topology and the Footmark Graph Method. We questioned the principle of suffix-optimal when finding MFP, and built up a precise urban topology model to validate our assumption. We generated Trajectory–Topology from origin GPS records data and smoothed the trajectories before building up the Footmark Graph. Furthermore, we split the Intersection Vertices into several Virtual Connection Vertices to eliminate the influence of transfer costs at intersections on MFP. The results show that our method can find more reasonable MFP than state-of-the-art baselines.

Future research could be improved in two aspects. Firstly, more efficient algorithms should be studied to find the most frequent path for large-scale datasets, to reduce time and space complexity. Additionally, when there are a lot of scarce data in the dataset, path analysis will be affected. Therefore, how to deal with sparse data to obtain effective most frequent paths is a problem that needs to be studied.

**Author Contributions:** Conceptualization, J.D. and Z.L.; methodology, J.D.; software, J.D.; validation, J.D. and X.J.; formal analysis, J.D. and X.J.; investigation, J.D.; resources, Z.L.; data curation, Z.L.; writing—original draft preparation, J.D.; writing—review and editing, J.D.; visualization, J.D.; supervision, Z.L.; project administration, X.J.; funding acquisition, X.J. All authors have read and agreed to the published version of the manuscript.

**Funding:** This research is partially supported by Shenzhen Project, China (JSGG20210802154807022), and NSF of Guangdong Province, China (2023A1515012716).

**Institutional Review Board Statement:** Not applicable.

**Informed Consent Statement:** Not applicable.

**Data Availability Statement:** The data that support the findings of this study are available from Hangzhou Transportation Satellite Positioning Application Co., Ltd. The data are not publicly available due to privacy restrictions.

**Conflicts of Interest:** The authors declare no conflict of interest.

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
