# Peer review of "Finding the Time-Period-Based Most Frequent Path from Trajectory–Topology"

_2504-2289, doi:10.3390/bdcc7020088_

Round 1

Reviewer 1 Report

Dear authors,

First and foremost, your English presentation of the work needs to be significantly improved. I am having hard time to understand importance of this work. Examined problem and its significance and application in modeling/planning practice needs to be better documented. Special emphasis needs to be provided on clear documentation of benefits of proposed approach as currently your results are very convoluted.

Author Response

Dear reviewer,

Thank you for your insightful comments on our article. We have revised the manuscript with the help of professional English language editors and corrected numerous grammar and spelling errors.

In addition, we have defined the research problem more clearly in the first section and strengthened the background introduction to enhance the paper's overall coherence.

Furthermore, we have improved the clarity of the research methods in the results section, highlighting the differences in our approach. We believe that these revisions have significantly improved the overall quality and readability of the manuscript.

Reviewer 2 Report

It would be helpful for the authors to provide a clear definition of the research questions in Section 1.

To provide a more comprehensive overview of the field, a systematic review of related work should be included.

Comparing the proposed method with other existing methods and highlighting its advantages would enhance the paper's contribution.

Including practical experimental results to compare the proposed method with other methods would further strengthen the study's findings.

It may be beneficial for the authors to discuss the potential limitations of the proposed method and outline future directions for this research in the concluding section.

Author Response

Dear reviewer,

Thank you for your insightful comments on our article. Here are our responses to your comments:

Point 1 & 2: It would be helpful for the authors to provide a clear definition of the research questions in Section 1. To provide a more comprehensive overview of the field, a systematic review of related work should be included.

Response: A new paragraph is added to Section 1 to provide a clear definition of TPMFP. In addition, we have defined the research problem more clearly in the first section and strengthened the background introduction to enhance the paper's overall coherence.

Point 3: Comparing the proposed method with other existing methods and highlighting its advantages would enhance the paper's contribution. Including practical experimental results to compare the proposed method with other methods would further strengthen the study's findings.

Response: Now we highlight that we compare our method with the state-of-the-art method in Section 6. Our method is used to find MFP(s) in comparison with MFP found by the other existing method.

Point 4: It may be beneficial for the authors to discuss the potential limitations of the proposed method and outline future directions for this research in the concluding section.

Response: Thanks for your advice. A new paragraph is added to the Conclusions to demonstrate the future directions for this research.

Reviewer 3 Report

The paper proposes an approach for Time Period-Based Most Frequent Path based on splitting the Intersections Vertices to eliminate the influence of transfer cost. The paper needs the following modifications:

1- Adding the contribution to the abstract as numerical summary 

2- adding recommendations and future work to the conclusion section

3- checking punctuation and grammar problems , line 178 space , line 180 (is kind a of ), line 226, 263 (presented) 

4- add explanation to the example in figure 5 caption

5- what is the time complexity of algorithm 1, we believe it can be reduced to make the algorithm more efficient ,, also for algorithm 2

6- explain figure 13, the meaning of not equal,, 

7- discuss these references 

Urban MV and LV Distribution Grid Topology Estimation via Group Lasso

Topology optimization for urban traffic sensor network

Clustering Point Process Based Network Topology Structure Constrained Urban Road Extraction From Remote Sensing Images

Modeling and Analysis of Large-Scale Urban Mobility for Green Transportation

Author Response

Dear reviewer,

Thank you for your insightful comments on our article. Here are our responses to your comments:

Point 1: Adding the contribution to the abstract as numerical summary.

Response: We have added the numerical summary to the abstract.

Point 2: Adding recommendations and future work to the conclusion section.

Response: A new paragraph is added to the Conclusions to demonstrate the future directions for this research.

Point 3: Checking punctuation and grammar problems, line 178 space, line 180 (is kind a of), line 226, 263 (presented) 

Response: Thank you very much for pointing out these mistakes!  We have corrected numerous grammar and spelling errors.

Point 4: Add explanation to the example in figure 5 caption.

Response: The explanation is added.

Point 5: What is the time complexity of algorithm 1, we believe it can be reduced to make the algorithm more efficient, also for algorithm 2.

Response: Now the time complexity of the two algorithms is discussed after they are proposed in the article.

Point 6: Explain figure 13, the meaning of not equal.

Response: We add "MFP(s) is not equal to MFP means that the transfer cost at intersections causes MFP to choose an incorrect path, as the example provided in Figure 12." to the Result section.

Point 7: Discuss these references.

Response: The references "Clustering Point Process Based Network Topology Structure Constrained Urban Road Extraction From Remote Sensing Images" and "Modeling and analysis of large-scale urban mobility for green transportation" is discussed in the eighth paragraph of Section 1. As for the reference "Urban MV and LV Distribution Grid Topology Estimation via Group Lasso", we are very sorry that we find it hard to establish a connection between it and our article, because it mainly discussed how to estimate the grid topology instead of using the topology structure.

Reviewer 4 Report

This paper studied the problem of finding time period-based most frequent 378 path based on Trajectory-Topology and Footmark Graph Method. The topic is interesting. I have some comments here.

1. The authors should pay attention to the grammar errors.

2. Did you consider familiarity in the paper?

3. The authors should add a table to show the difference between their method and other methods.

4. Did you consider different weather conditions?

Author Response

Dear reviewer,

Thank you for your insightful comments on our article. Here are our responses to your comments:

Point 1: The authors should pay attention to the grammar errors.

Response: We have revised the manuscript with the help of professional English language editors to correct numerous grammar and spelling errors.

Point 2: Did you consider familiarity in the paper?

Response: We have defined the research problem more clearly in the first section and strengthened the background introduction to enhance the paper's overall coherence.

Point 3: The authors should add a table to show the difference between their method and other methods.

Response: The main difference between our method and other methods is that our method eliminates the transfer cost. Now we highlight that we compare our method with the state-of-the-art method in Section 6. Our method is used to find MFP(s) in comparison with MFP found by the other existing method. The results are shown in Figure 13, we are very sorry that we find it hard to use a table to describe the difference.

Point 4: Did you consider different weather conditions?

Response: From our point of view, our method is to study the travelling behaviors of drivers in a specific period. Thus, the influence of weather conditions is contained in the behaviors of drivers during the period.

Round 2

Reviewer 2 Report

The authors have graciously taken into consideration my review comments and have made revisions to the manuscript accordingly. Based on the improvements made, I highly recommend accepting the submission.